# Solvation of Model Biomolecules in Choline-Aminoate Ionic Liquids: A Computational Simulation Using Polarizable Force Fields

**DOI:** 10.3390/molecules29071524

**Published:** 2024-03-28

**Authors:** Stefano Russo, Enrico Bodo

**Affiliations:** Chemistry Department, University of Rome “La Sapienza”, P.le Aldo Moro 5, 00185 Rome, Italy; stefano.russo@uniroma1.it

**Keywords:** polarizable force fields, molecular dynamics, ionic liquids, biomolecules

## Abstract

One can foresee a very near future where ionic liquids will be used in applications such as biomolecular chemistry or medicine. The molecular details of their interaction with biological matter, however, are difficult to investigate due to the vast number of combinations of both the biological systems and the variety of possible liquids. Here, we provide a computational study aimed at understanding the interaction of a special class of biocompatible ionic liquids (choline-aminoate) with two model biological systems: an oligopeptide and an oligonucleotide. We employed molecular dynamics with a polarizable force field. Our results are in line with previous experimental and computational evidence on analogous systems and show how these biocompatible ionic liquids, in their pure form, act as gentle solvents for protein structures while simultaneously destabilizing DNA structure.

## 1. Introduction

The interaction of ionic liquids (ILs) with biological matter [1], especially natural polymers, has attracted attention in recent years due to the possibility of exploiting the peculiar solvation properties of these substances in medicine applications [2,3,4]. Despite the toxicity of several variants of ILs, refs. [5,6,7,8] the possibility of tailoring their molecular constituents to specific purposes has spawned a variety of specialized classes of ILs that are biocompatible and suited for their use in living organisms [9].

Among the various attempts, the most promising category of biocompatible ILs is likely to be found among the so-called protic ILs [10], which are essentially the result of an acid-base reaction (albeit sometimes only formally) where the proton transfer generates the charge separated species—a protonated base and a deprotonated acid. More specifically, it is possible to obtain a vast class of biocompatible ILs by coupling the cholinium cation with various forms of organic acids and amino acids [11,12,13]. ILs based on cholinium and a deprotonated amino acid ([Ch][AA] ILs) are biocompatible [14,15,16,17,18,19] and easy to synthesize from inexpensive ingredients [20].

Despite a notable experimental effort in providing data about the compatibility of ILs with biological matter, biopolymers, and biomolecules in general, refs. [21,22,23,24,25,26,27,28,29] the fundamental (i.e., molecular) mechanisms by which the IL interacts with biomolecules is not yet entirely understood (see for example [30] and references therein). This is due to an intrinsic difficulty in measuring the relevant molecular observable in complex systems [31,32,33] where disentangling the interactions of biomolecules (which are inherently complex chemical systems) and solvent ILs is often unpractical. Furthermore, molecular simulations of biomolecules in ILs are time-consuming, system specific, and often suffer from technical difficulties due to the availability of suitable and reliable force fields [30,34,35]. Despite these problems, several attempts have already been made focusing on imidazolium-based ILs [36,37,38,39,40]. A significantly smaller number of studies are found for cholinium- and/or aminoacid-based ILs. These are mainly focused on the stability of model proteins and ribonucleotidic systems in IL/water mixtures. Sundaram et al. [41] studied the stability of insulin in cholinium glycinate [Ch][Gly], alaninate [Ch][Ala], and prolinate [Ch][Pro] at different concentrations in water. They noticed that water was displaced from the first protein solvation shell and that the solvation of the macromolecule was dominated by the anion with prolinate essentially preserving the crystal structure from the denaturising action of water. Chevrot et al. [42] also showed that amino acid anions solvated small protein structures displacing water from the inner shell. Bisht et al. [43] argued that an amino acid anion such as glycinate was more disruptive toward bromelain structure than bromine using experiments and docking models. Other recent studies focused on the stability of nucleic acids structures in water/IL solutions [29,44,45]. Most of the data pertains to traditional ILs, refs. [46,47,48] long DNA double strands (ds), and IL/water mixtures [28,49,50].

Following the suggestions in [35], we present here an analysis of the solvation of two very small model biomolecules in three different neat [Ch][AA] ILs using a polarizable force field. It has been shown in [51,52] that in concentrated electrolytes such as ILs, the traditional, charge-fixed MD models, although still widely used, overestimate friction and solvation strengths. To remedy this intrinsic deficiency, many simulations of ILs are carried out using a scaled-charge approach where the scaling parameter is optimized for a specific system and realigns the computed dynamic properties with real observables [52,53]. It is our opinion that instead of using an adjustable parameter, it would be desirable to employ a true polarizable force field to model the balance of the long-range interactions (i.e., dispersion and electrostatic). In addition, it is becoming clearer that the introduction of explicit polarization in force fields for biomolecules may improve the quality of the results owing to a more realistic treatment of the electrostatic interactions [54,55,56].

We have recently developed a fully polarizable force field for [Ch][AA] ionic liquids including nine different amino acid anions representative of the different families, from the simple [Gly], to aromatic [Phe], acid [Asp], and basic ones [Lys] [57]. The force field was validated to reproduce the short-range coordination properties of anions and cations in the pure liquid phase of the ILs. The force field is based on a multipolar expansion of the electrostatic energy that includes first order induced dipoles. It is built out of the *AMOEBA* model [58].

In this work we combine our newly developed polarizable force field together with the well-established polarizable counterparts for bio-matter *AMOEBA-nuc* and *-pro* [59,60,61] to analyze the behaviour of model biomolecules immersed in neat [Ch][AA] ILs. We will present structural data with the aim of elucidating the molecular features of the biomolecule–ILs interaction and, provide a minimal basis for an accurate nanoscopic interpretation of the many and variegated results obtained in the laboratories.

As test case studies, we have chosen two simple biological systems: an oligopeptide and a ds-oligonucleotide. Since the calculations are computationally demanding, we decided to limit the size of the system and use the 12-residue peptide (KTWNPATGKWTE, PDB id #2EVQ), which has a β-hairpin shape and appears a good candidate to check how well the folded structure is preserved in the IL. For DNA, we chose the 1LJX structure, such as the hexanucleotide 5′-TGCGCA-3′. It is the shortest DNA sequence that already exhibits the double helix conformation. The structures of both biomolecules are reported in Figure 1.

## 2. Results and Discussion

### 2.1. 12-Residue Oligopetide

We begin our analysis of the results with the oligopeptide model. Figure 2 reports the RMSD(t) function with respect to the initial NMR experimental structure along the first 5 ns of simulation time. The time span reported in Figure 2 is representative of the equilibrated systems. Due to our methodological setup in the equilibration phase (see Section 3), the initial displacement from the reference structure is very fast (within few ps) and essentially appears as a vertical line at t = 0.

For all four of the systems ([Ch][Gly], [Ch][Ser], [Ch][Lys], and water), most of the conformational changes of the protein takes place in the very first few ps of dynamical evolution, after which the deviation from the initial structure stabilizes and reaches a plateau. We notice that the structure in water shows the largest conformational change and the most significant fluctuations. This is due partially to the force field which is different from the one used to optimize the NMR-deposited structure but, foremost, to the high mobility of the two terminal residues Glu and Lys (see also the analogous results obtained using polarizable models in [62]).

Remarkably, the action of the three ILs is gentler than water, and the protein structure presents a minor extent of structural modifications with the RMSD remaining confined below 2 Å.

A more precise idea of the structure variations can be had using the data in Appendix A where we report the R factor [63] in the four solvents. The quantity R is the sum of the ratios of the distance between two corresponding residues on opposite β-strands and its value in the reference structure. If the hairpin maintains its folded state, these ratios are almost one and R is essentially equal to the number of distances chosen. In Appendix A we report the R factor as obtained from four distances starting with the two terminals residues (r1) and proceeding upward toward the turn (r2, r3, and r4). The four residues of the turn are not considered. In all three ILs, the R factor is practically four, hence providing evidence that the system largely maintains its folded state with very limited conformational evolution with respect to the native state. These findings appear to be in agreement with previous findings on other IL containing aminoate anions [42]. The data pertaining to water show a significant deviation of R from four towards larger values. A closer inspection of the individual distance ratios, however, makes it apparent that most of the deviation from the reference structure (and to the RMSD of Figure 2) is due to the two terminal residues whose positions fluctuate greatly in water.

Data in Figure 3 pertain to the [Ch][Gly] IL and have been obtained by counting the average number of molecular ions in direct proximity of a given residue (x-axis). The solvation of the protein in this IL is driven mainly by the glycinate amino acid anion that is found to lie in proximity of all residues, albeit with a preference for the Lys-1 terminal and the adjacent Thr-2 of the β-strand. The cholinium cation has instead an uneven distribution in the immediate vicinity of the protein structure with a preference for the two β-strands and the two aliphatic residues in the turn (Ala and Gly). The cation seems to remain far from the strongly hydrophilic Asn and from the Pro and Thr of the turn.

A more comprehensive view of the relative positions of the two IL ions and every residue of the protein is shown in Appendix A where we report the radial distribution functions between the centres of mass of the molecular ions and the 12 residues of the hairpin. For all residues, with the exception of Gly-8, the amino acid anion is the closest solvating molecular ion.

The stability of the folded structure in the [Ch][Gly] structure is achieved because of the peculiar solvation at the N- and C-terminal residues at the end of the two strands. At the difference with water, the IL is unable to break the Lys–Glu H-bond where the Lys ammino group is protonated and binds to the Glu carboxylate.

A representative structure extracted from the simulation is shown in Figure 4. We show (with sticks) the two ends of the protein (Lys and Glu) and two nearby IL molecular ions (spheres). The interaction between the protein and the IL is not able to completely disrupt the H-bond and the proximity of the two terminal residues is helped by the glycinate anion that puts itself in a bridge position. The cholinium remains H-bonded to the glycinate, however it has a mere spectator role.

The solvation of the hairpin structure in the [Ch][Lys] IL is illustrated in Figure 5. The data are presented in the same fashion as Figure 3 and show a similar pattern where the lysinate surrounds the protein structure with the cholinium being less interacting. However, no simple pattern about the preferences of either the cholinium and the aminoate anion seem to emerge clearly from the comparison of the [Ch][Gly] and [Ch][Lys] solvation, apart from the mentioned preference of the anion to stick closer to the peptide. In general, it seems that with lysinate, cholinium is more abundant in the first solvation shell and solvates all residues with the exception of Lys-9.

The folded structure is stabilized by an H-bond between the terminal Lys and Glu. Lysinate appears to form an additional H-bond with the Lys residue, but its interaction is not sufficient to produce any structural disruption in the protein terminals. A typical arrangement in the terminal region of the protein is shown in Figure 6 in the same fashion as Figure 4. A more thorough view of the individual solvation shells of the 12 residues can be had from the data of Appendix A, where we report the radial distribution functions of the cations/anions with the residues.

Solvation by the [Ch][Ser] does not show significant differences with respect to the previous two ILs, and it is still dominated by the anion while the cation seems to interact less with each residue. The situation is illustrated by Figure 7 in a similar fashion as before. The serinate carboxylate binds the terminal Lys through H-bond, while the cholinium often remains in the immediate proximity of the Trp residues. Again, the serinate establishes a close contact with most of the residues, especially the terminal ones (Lys-1 and Thr-2), while cholinium is found in proximity to Trp-3 and Thr-7 and Thr-11 bound by hydrophobic interactions. A typical arrangement is reported in Figure 8. Overall, the OH group in the serinate does not appear to play a particular role.

### 2.2. Six Base Pairs DNA Double Strand

Our chosen model is a six-base-pair oligonucleotide in B conformation. It is well-known [64] that such short oligonucleotides are less stable than longer ones, and that they are stabilized in water solution by high ionic strength where buffer ions neutralize the phosphate groups (in our water simulation we used Mg^2+^ ions). However, the system is intrinsically prone to melting and denaturation processes.

In this context, this is an advantage because it makes possible the observation of dynamic processes without the need for extremely long simulation times that would have been prohibitive with polarizable force fields.

The ds-oligonucleotide behaviour in the water and the three ILs is illustrated in Figure 9. There, we report the RMSD(t) as a function of time along the initial five nanoseconds of simulation common to all simulations. Again, in this case, the reported time section is representative of the equilibrated system. As for the peptide, the structural change with respect to the initial (crystallographic) geometry takes place in the very first few ps of simulation as soon as the constraints on the biomolecule are removed. Then, it reaches a plateau. Contrary to what we have seen for the oligopeptide, water in this case is the solvent that mostly preserves the crystal structure, while the ILs have a pronounced effect on it depending on the anion identity. Lysinate behaves similarly to water in preserving the overall structural features of the DNA complex. Serinate significantly alters the structure, and glycinate has a dramatic effect with an almost complete disruption of the double helix.

A more pictorial idea of the situation at the end of simulation can be had by looking at Appendix A where we plotted the final structure of the DNA model after the dynamics. It is clear that the IL has a strong effect on the structure of the DNA complex and that it can promote a fast and efficient denaturation. Whether this is a universal behaviour of these kind of ILs is a conclusion that remains foreign to us due to the limited timeframe of our simulations. We certainly see that [Ch][AA] ILs with certain aminoate anions (glycinate and serinate) exert a partial or global disruptive effect on the H-bonded base parings in a relative short time.

Among the three ILs, the only one that does not lead to appreciable disruption of the base pairings in the DNA structure within the chosen time frame is [Ch][Lys] (grey line in Figure 9). Also, as already noted with the polypeptide, the structural fluctuations are suppressed with respect to water, probably due to the much higher viscosity of the IL. For [Ch][Lys], the N-N distances of the four G-C and two T-A H-bonded pairs remained very similar to the values attained in water, and all of the six base pairs lasted essentially intact for fourteen nanoseconds.

Contrarily to what we have seen for the hairpin protein, the most abundant molecular ion in the first solvation shell of the DNA model is cholinium, an effect obviously induced by the negatively charged phosphate groups. This agrees with previous simulations and experimental results for these kind of liquids that have been found in the case of IL/water mixtures [65,66,67].

A quantitative assessment of the composition of the solvent layer around the DNA structure is reported in Figure 10 for [Ch][Lys] and in Appendix A for all ILs. The histograms correspond to the average count of cations (blue) and anions (orange) within 7 Å of the c.o.m. of specified base (nitrogenous base only, excluding phosphate and ribose).

We can clearly see that for [Ch][Lys] (and for [Ch][Ser], but to a lesser extent) the cation dominates at short distances from the base. Only the most exposed terminal bases interact with the anions. This is due to two effects: (1) the cation is attracted in proximity of the bases by the negatively charged phosphate groups and (2) the size of the anion has a relevant effect and the bulky chains of lysinate provides an unfavourable steric effect.

A more detailed measure of the interaction between the components of the liquid and the phosphate groups is shown in Appendix A (top panels) where we report few radial distributions functions between the PO_4_ group and selected atoms of the molecular ions. For lysinate (top panels, Appendix A), the cation binds to the phosphate via favourable electrostatic (with a peak around 5 Å in the PO_4_-N distance) assisted by the formation of an H-bond between the negatively charged PO oxygen and the OH group (with a peak at 4 Å in the PO_4_-OH distance). For lysinate, the interaction between PO_4_ and the anion is negligible, with only residual interactions at distances around 3 Å when mediated by H-bonds with the amino group.

The structural features of the IL surrounding the DNA strands are similar when we use serinate instead of lysinate, although the former does not preserve the DNA structure. As for Lys and Ser, the cholinium is the nearest molecular ion in the solvation shell (Appendix A, top-left panel). However, an increased proximity of the serinate anion near the two ends of the DNA sequence can be clearly noticed and is likely due to the smaller size of Ser with respect to Lys. The data in Appendix A (middle panels) further confirm that the interaction between the phosphate groups and the cholinium is similar (if not slightly stronger) for the serinate-based IL when compared to [Ch][Lys]. At the same time in [Ch][Ser] the interaction between phosphate and the amino group of the aminoate is more pronounced than in lysinate with a clear peak due to H-bond with PO_4_-N distances of the order of 3 Å (Appendix A, left-middle panel). We also notice a significant contribution due to the OH group of serinate with PO_4_-OH distances around 4 Å.

As an example of the cholinium interaction with phosphate groups, we report in Figure 11 the final snapshot of the DNA model in [Ch][Ser]. The two terminal phosphate groups are displayed as yellow van der Waals spheres, while the surrounding cations are shown in blue.

As shown by the data in Figure 9 (and by the structure of Figure 11), the structural deviation with respect to the crystal structure, in [Ch][Ser], is significant and around 4 Å. The simulations for [Ch][Ser] lasted for about 10 ns. During this time, we saw that while five out of six base pairs survived, one of the two terminal AT pairs was significantly perturbed by the IL and was effectively broken apart (see Appendix A). According to our simulations, the driving force beyond the base pair disruption in [Ch][Ser] is the weakening of the H-bond contact between adenine and thymine and a progressive solvation by both the cation and the anion of the free nitrogenous base. The initial event leading to the H-bond breaking seems to be linked to the interaction of the serinate with the exposed bases at one of the two ends of the duplex that weakens the H-bond, perturbs the stacking, and leads to a destabilization of the secondary structure. When the bridge between the two bases is broken, both cations and anions partake to the solvation of adenine and thymine with the cation having a slight prevalence at short distance. Both molecular ions use their –OH group as the main anchor to the aromatic structures, but the resulting H-bonds are very weak with large acceptor donor distances (≥3 Å). We show an example of the final situation in Figure 12, where the two free nitrogenous bases have been highlighted using van der Waals spheres and the surrounding molecular ions are color-coded: blue for cations, red for anions.

Solvation by anions is much more prominent in the glycinate compound (Appendix A), but the anion proximity to the bases, in this case, is post-denaturation, i.e., an inevitable result of the almost complete breaking of the DNA structure. This disruption of the DNA sequence is very fast and takes place within the first six nanoseconds of simulation. At that point, the nitrogenous bases are exposed to the IL. The solvation of the broken DNA is due to both cations (near phosphate and to a lesser extent ribose) and anions which are prevalent near the exposed bases. A snapshot of broken DNA sequence along with few selected solvating molecular ions is in Figure 13 (anions in red, cations in blue). It is very likely that the affinity of the glycinate for the nitrogenous bases and its small size play an important role in determining the fate of small DNA oligonucleotide.

## 3. Methods and Computational Details

The details of the *AMOEBA* polarizable force field for the ILs have been presented in [57] together with its validation by comparison with ab initio data. The parameters for the biomolecules have been taken from the most recent *AMOEBA*-*pro* and -*nuc* parametrizations [59,61].

Validation of the force field has already been presented in great detail [57]. The compatibility of our parameters in combination with *AMOEBA-pro* and -*nuc* is presented in detail (Appendix A). For the 2EQV oligopeptide, the reference structure is the NMR structure [68] in water, while for the ds-DNA model (d(TGCGCA)), we used the X-ray crystallographic structure [69].

In both cases, we created three cubic boxes containing the macromolecule and five-hundred ionic couples of [Ch][Ser], [Ch][Lys], [Ch][Gly]. In addition, and for reference purposes, a fourth box with the biomolecule and nine-thousand water molecules was also created. For the latter, Cl^−^ (for the protein) and Mg^2+^ (for the DNA) have been used to neutralize the cell, while for the ILs a slight excess of cations or anions was inserted for the same purpose.

The cells were created by placing the solvent molecules around the macromolecule at low density. The solvent was then equilibrated under NPT conditions (P = 1 bar, T = 300 K) with the biomolecule structure kept frozen. When the system reached equilibrium, the constraints on the peptide are removed, and the final simulations NVT (T = 300 K) are produced.

The time propagation has been conducted using the respa algorithm with a timestep of 0.5–1 fs (depending on the system). The trajectory was saved every 0.1 ps and the overall length of the NVT simulations ranges from a minimum of 5 ns to a maximum of 15 ns depending on the system (see Appendix A for individual details). Some simulation times are longer depending on the nature of the physical process taking place during dynamics. For example, a significant alteration to the structures may imply a shorter simulation or a longer one depending on whether the changes occurred rapidly or not. Temperature control was achieved using the Bussi–Parrinello thermostat with a constant of 0.1 ps, while pressure control using Berendsen barostat with a constant of 2 ps. Both van der Waals and electrostatic cutoffs were set to 10 Å. Particle mesh Ewald was used to compute electrostatic energy at long range. All dynamics have been performed using the Tinker-HP software [70].

As several authors pointed out [71], stability of protein or other biomolecules in ILs could require the sampling of timescales long-enough to account for the high viscosity of the surrounding IL. Our simulations might be too short (5–14 ns) to expose slow dynamical process leading to denaturation, but they nevertheless provide useful insights into the short timescale dynamics and into the structural properties of the ILs surrounding the biomolecule. Moreover, as we shall see, for certain systems analysed here, only a few ns were sufficient to observe important dynamical processes. Overall, the entire set of simulations presented here employed nearly six million core-hours on a cluster based on AMD Zen 2 EPYC processors with an average concurrent use of 128 cores.

## 4. Conclusions

In this work we have reported the results obtained by simulating model biomolecules immersed in neat bio compatible [Ch][AA] ILs. Different from many other previous simulations of biomolecules in ILs, our approach is based on a fully polarizable force field, hence we expect an accurate prediction of the resulting structure. Due to the nature of the force field and the constrains on the number of atoms (solvation of biomolecule requires many solvent molecules), computational overhead is inevitable, and we were able to sample only short time scales between 10–15 ns. Nevertheless, we were also able to grasp some important dynamical phenomena taking place in those limited times.

The first model is a small 12-residue hairpin protein. The solvating ILs (three different anions) preserved the folded state of the protein more effectively and with less structural fluctuations with respect to water. Interaction of the ILs with the backbone was similar for the three ILs and was characterized by a prevalence of aminoate anions in the first layer of solvation. The aminoate provided various H-bond motifs with the terminal residues but was nevertheless unable to induce any breaking in the secondary structure. These findings agree with data from existing literature and previous models.

The second model is the shortest double helix DNA duplex. In this case, the effect of the surrounding IL was more diverse, strongly dependent on the anion, and more pronounced than in the protein case. The simulations reported here tell that the glycinate has a strong destabilizing effect on the DNA duplex and effectively breaks the base H-bond pairing quite efficiently on a timescale of a few ns. A larger anion such as serinate, due also to their polar side chain group, acted more slowly. Nevertheless, the simulation indicated a propensity toward a partial breaking of the base pairs. Lysinate, owing to its bulky side chains, did not appreciably compromise the DNA structure in the time span of our simulations. Solvation of the DNA duplex was dominated at short range by cholinium cations that bind the phosphate moieties. Lysinate anions, with its bulky chains, was the least effective in solvating the DNA and the latter remained essentially intact and with structural features similar to those in water throughout the 14 ns simulation. We note that in the neat [Ch][Lys] liquid (used in our simulations), the amino group is not protonated. Serinate, due to the presence of a polar side chain (–OH), was more involved in solvating the most exposed nitrogenous bases whose pairing was partially broken during the simulation in 10 ns. The weakening of the stacking and of the intra-base H-bond induced the breaking of one of the AT pair, and was followed by an increasing anionic solvation of the resulting bare nitrogenous bases. Although no other subsequent base pair disruptions have been recorded in our simulation, this AT breaking seemed to point to an initial stage of a general instability of short DNA oligomers in [Ch][AA] ILs. This would also be in accord with the evidence gathered by using the IL based on the glycinate anion whereby a complete separation of the two strands of DNA was recorded in just less than five nanoseconds of simulation. In particular, the small glycinate anion owing to its small size resulted to be very effective in disrupting the secondary structure of the DNA model.

## Figures and Tables

**Figure 1 molecules-29-01524-f001:**
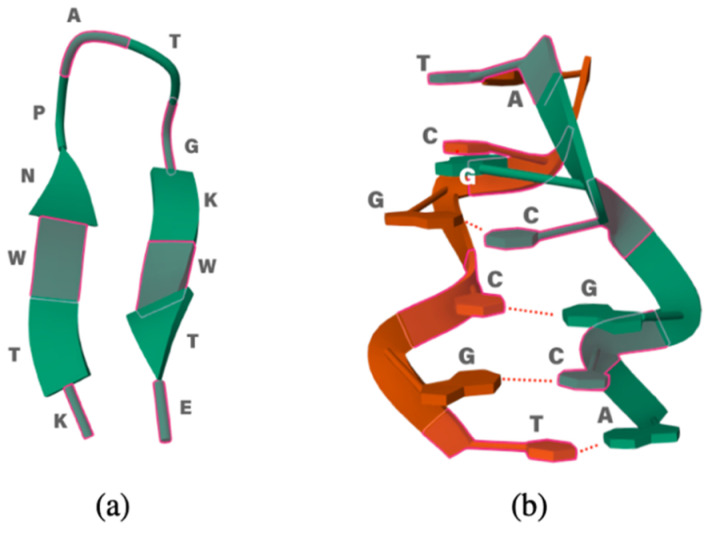
(**a**) Small polypeptide 2EVQ and (**b**) 1LJX DNA model.

**Figure 2 molecules-29-01524-f002:**
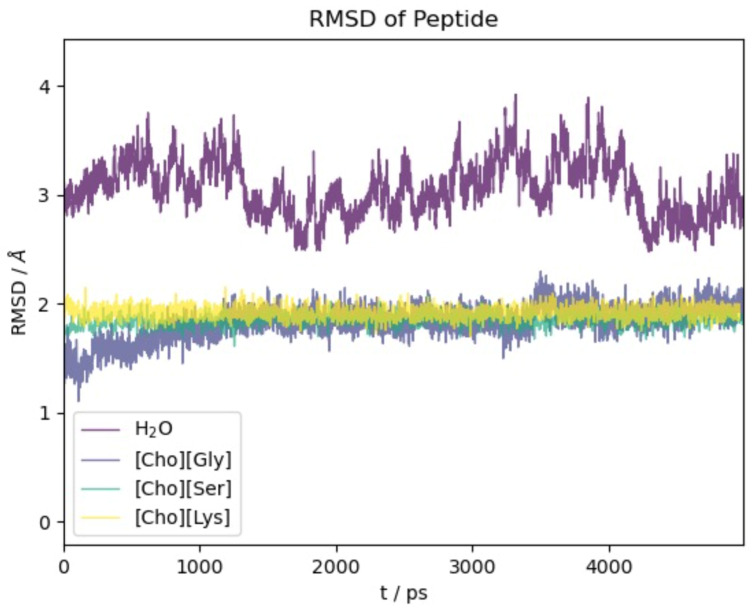
RMSD with respect to initial experimental structure as a function of time for 2EVQ oligopeptide in different solvents.

**Figure 3 molecules-29-01524-f003:**
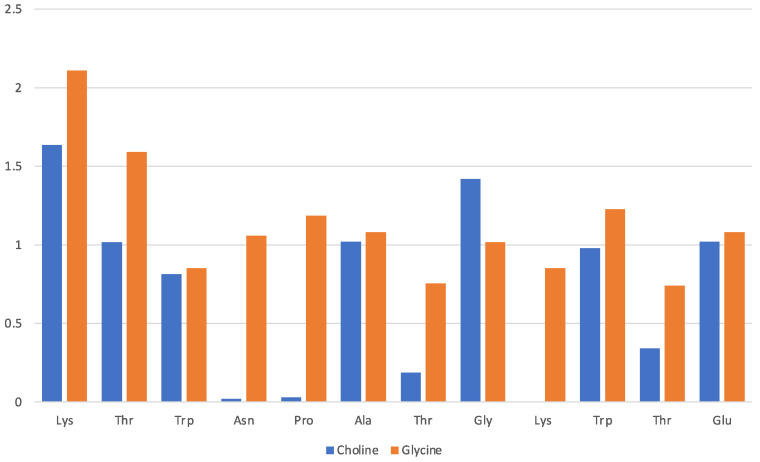
[Ch][Gly] liquid with oligopeptide: average number of solvating molecular ions per residue. Solvating ion is counted if its c.o.m lies within 6 Å from residue.

**Figure 4 molecules-29-01524-f004:**
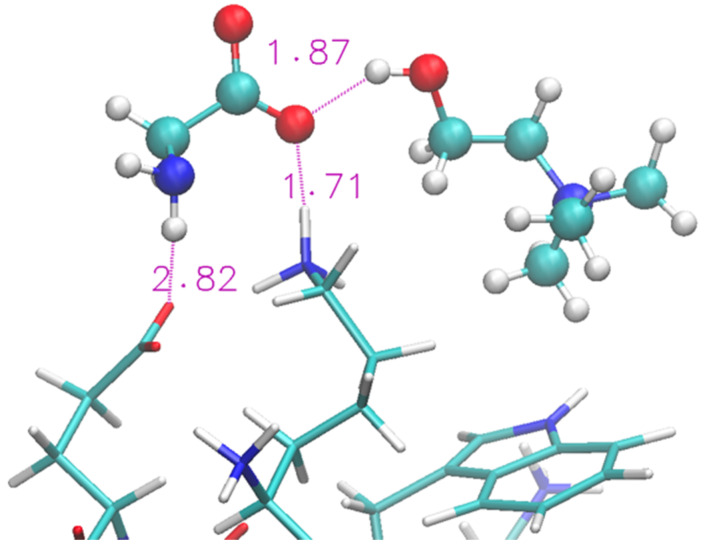
Zoom of terminal residues of oligopeptide in [Ch][Gly]. Proton-acceptor distances are indicated to highlight H-bonds between protein and liquid.

**Figure 5 molecules-29-01524-f005:**
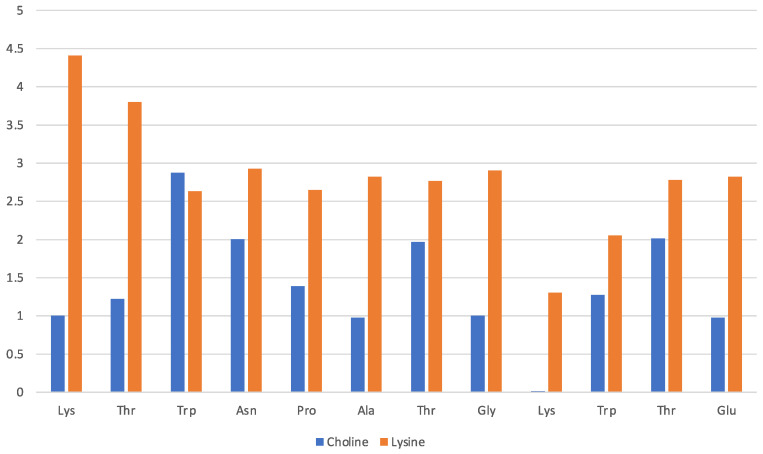
[Ch][Lys] liquid with oligopeptide: average number of solvating molecular ions per residue. Solvating ion is counted if its c.o.m lies within 7 Å from residue.

**Figure 6 molecules-29-01524-f006:**
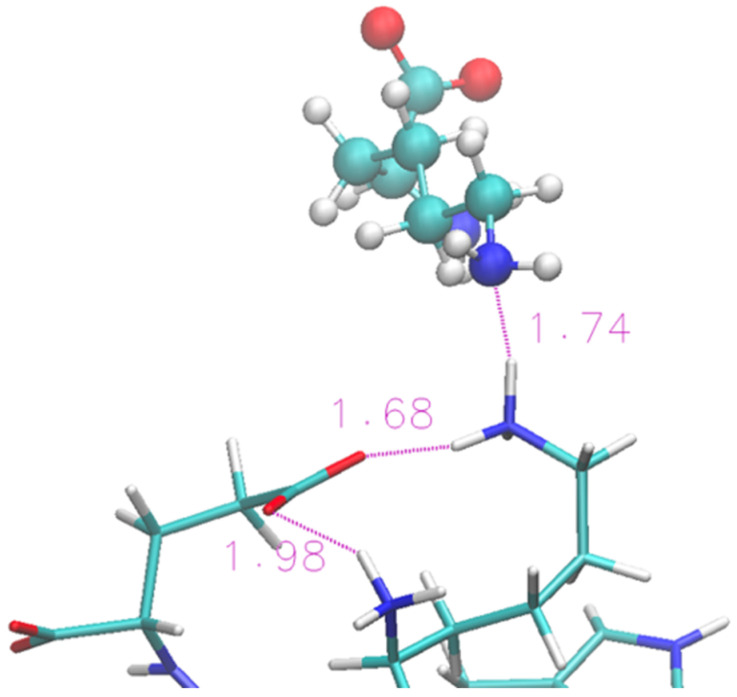
Zoom of terminal residues of oligopeptide in [Ch][Lys]. Proton-acceptor distances are indicated to highlight the intra-protein H-bonds and an occurrence with lysinate ion of IL.

**Figure 7 molecules-29-01524-f007:**
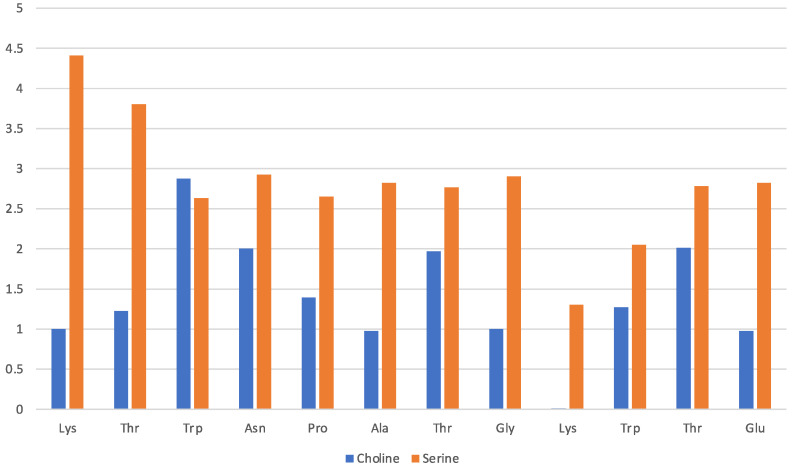
[Ch][Ser] liquid with oligopeptide: average number of solvating molecular ions per residue. Solvating ion is counted if its c.o.m lies within 7 Å from residue.

**Figure 8 molecules-29-01524-f008:**
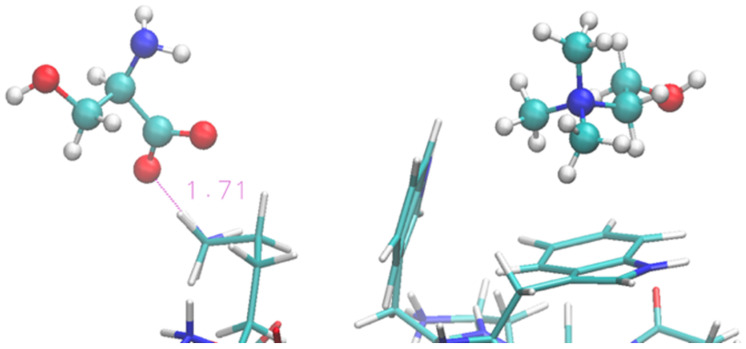
Zoom of terminal and middle residues of oligopeptide in [Ch][Ser]. Terminal Lys coordinated by serinate can be seen on the left, while two Trp of the strand are on the right in interaction with cholinium.

**Figure 9 molecules-29-01524-f009:**
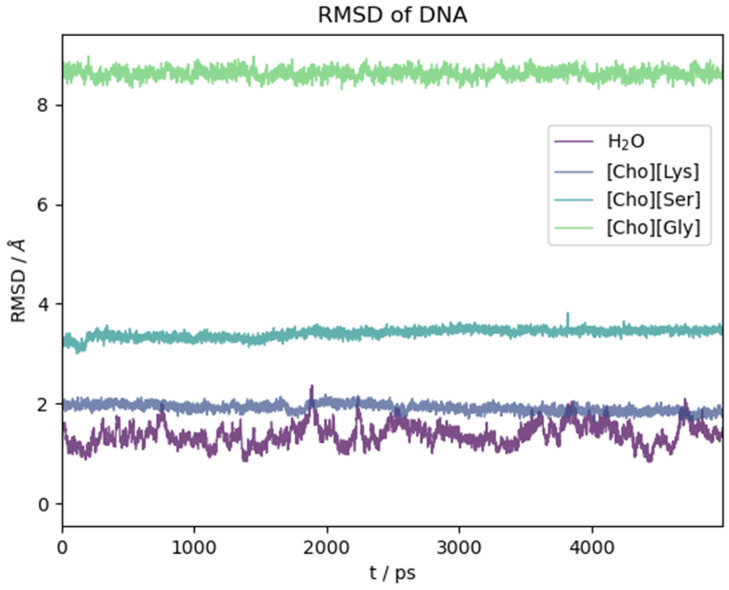
RMSD with respect to initial crystallographic structure as a function of time for DNA model in different solvents.

**Figure 10 molecules-29-01524-f010:**
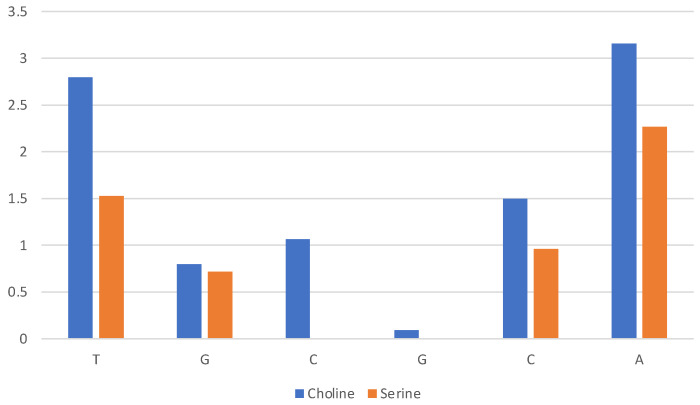
Composition of solvent layer surroundings of DNA model in [Ch][Lys]. Histogram reports average number of cations (blue) and anions (orange) within 7 Å of c.o.m. of base (excluding phosphate and ribose).

**Figure 11 molecules-29-01524-f011:**
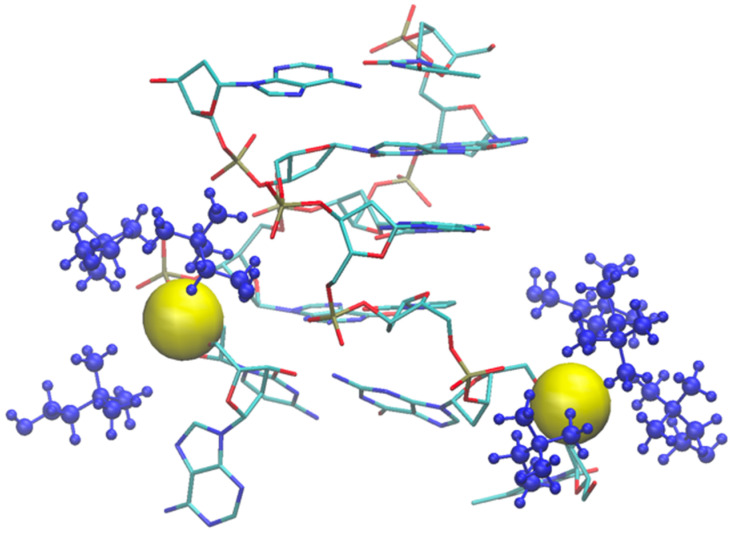
Final snapshot of DNA model in [Ch][Ser] where one AT base pair broke. Two involved phosphate groups are shown as yellow van der Waals spheres, and coordinating cholinium cations are blue.

**Figure 12 molecules-29-01524-f012:**
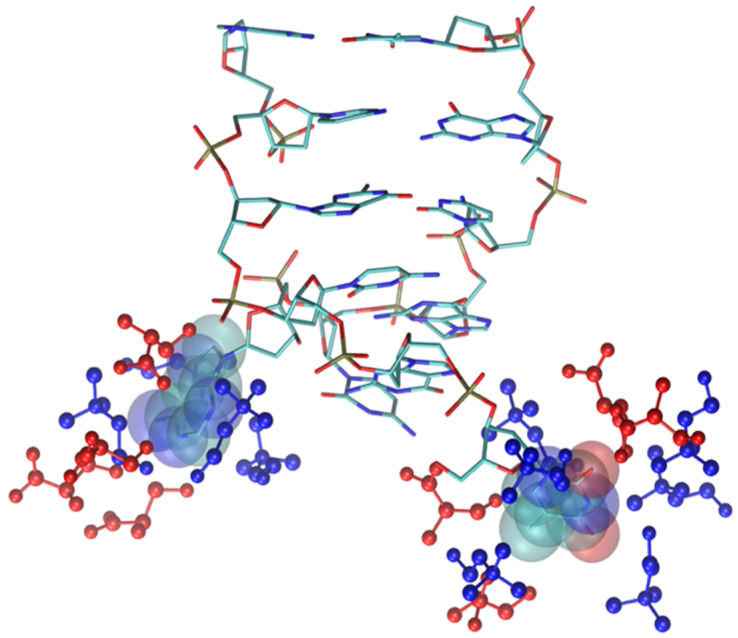
Final snapshot of DNA model in [Ch][Ser]. Two nitrogenous bases whose H-bond has been broken are displayed with van der Waals spheres, IL molecular ions are displayed in blue (cations) and red (anions). H atoms have been suppressed.

**Figure 13 molecules-29-01524-f013:**
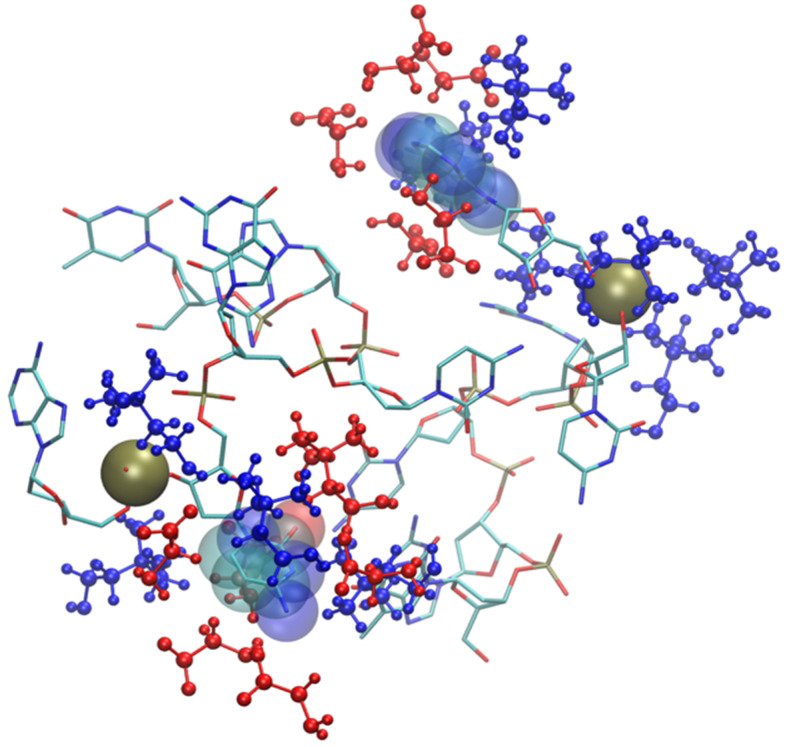
Final snapshot of DNA model in [Ch][Gly]. Two exposed nitrogenous bases are displayed with van der Waals spheres, IL molecular ions are displayed in blue (cations) and red (anions). Phosphate groups are rendered as yellow spheres. H atoms have been suppressed.

## Data Availability

The raw data supporting the conclusions of this article will be made available by the authors upon reasonable request.

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
