# Peer review of "Solvation of Model Biomolecules in Choline-Aminoate Ionic Liquids: A Computational Simulation Using Polarizable Force Fields"

_molecules, 2024, doi:10.3390/molecules29071524_

Round 1

Reviewer 1 Report

Comments and Suggestions for Authors

Comments: minor revisions needed as noted

The study of Russo and Bodo is devoted to the analysis of the behavior of model biomolecules immersed in neat ionic liquids using fully polarizable force field. By choosing two rather common biomolecules, the authors show that a AMOEBA force field enables an accurate prediction of the resulting structure. As a drawback, the total simulation time is small and does not exceeds 15 ns. At the same time, the amount of new results for chosen systems is already sufficient for publication. Overall, I would recommend accepting this paper for publication after a couple of technical improvements.

Namely, please provide the exact simulations times and the amount of computational resourced spent for each systems in the SI. In addition, please indicate the software package in which the simulations have been performed. Finally, it would be great to assess the maximum scale of the potential simulation system, which could be simulated by AMOEBA force field in the nearest future regarding the current availability of computational resources.

Indeed, the authors may argue that providing such information is not necessary since it can be found in the respective references. On the other hand, including the technical information in the open access paper will guarantee the authors the gratitude from the relevant research communities. Besides, along with scientific and technical importance, such information might be useful in the routine actions, for instance, when applying for computation time in various supercomputers.

Author Response

We thank the reviewer for their positive assessment of our paper. Here are the required answers and a list of actions taken. 

Namely, please provide the exact simulations times and the amount of computational resourced spent for each systems in the SI. In addition, please

We have added a table specifying these data in the SI in a new section S5. We have also added a short text to point to these details in the methods section.  

indicate the software package in which the simulations have been performed. 

We have now provided this information in the text with its reference (n. [70]). 

Finally, it would be great to assess the maximum scale of the potential simulation system, which could be simulated by AMOEBA force field in the nearest future regarding the current availability of computational resources. Indeed, the authors may argue that providing such information is not necessary since it can be found in the respective references. On the other hand, including the technical information in the open access paper will guarantee the authors the gratitude from the relevant research communities. Besides, along with scientific and technical importance, such information might be useful in the routine actions, for instance, when applying for computation time in various supercomputers.

We do not have a simple general answer to this question. The current results have "eaten" several months and several millions of core hours on a Euro-HPC cluster (Karolina). We have added a sentence at the end of the methods section to report these numbers. 

Reviewer 2 Report

Comments and Suggestions for Authors

This manuscript reports a molecular dynamics study of two biological molecules in three types of ionic liquids. In addition to providing a comprehensive description of molecular interactions between the biomolecules and ionic liquids, the authors compared the conformational changes in ionic liquids to MD simulations in water. The work presented here shows great potential for further research and practical applications.

While the paper is very well written and MD simulation reveal interesting results, I have a few comments and questions:

  1. The MD simulations of the two biomolecules generated significantly different results; the oligopeptide maintained its structure in ionic liquids while the DNA fragment did not. Furthermore, one of the three ionic liquids denatured the oligonucleotide duplex. However, both figures 2 and 9 and show little variation of RMSD values during the 5 ns. If the equilibration step was done with frozen atomic coordinates, one would expect to see the original structure at time t=0. If both illustrations show the last 5 ns of each MD simulation, the illustrations should be corrected to accurately reflect the actual time, although the entire MD run should be presented and not only the last 5 ns.

  1. Figure 9 caption includes "Figure 1", probably from an earlier version.

  1. Out of 8 MD simulations, some were 15 ns runs while others were shorter, however it is not clear how long was each distinct simulation. This information is especially important and it should be included in the Methods section.

  1. On one hand, the initial structure of the oligopeptide is based on NMR, which could account for the lesser variability during the MD simulation. On the other hand, the initial structure of the DNA fragment is based on X-ray and I think it would be worthwhile to compare the NMR and X-ray pdb structures of similar DNA fragments reported in the database. What is the RMSD between these structures?

  1. One final suggestion: In the conclusions section, the last sentence should be reworded. I understand its meaning, but a minor change could elevate the significance of this work.

Author Response

We thank the reviewer for assessing our paper and for the useful suggestions. Here is a point-by-point reply to their requests for clarifications. 

  1. The MD simulations of the two biomolecules generated significantly different results; the oligopeptide maintained its structure in ionic liquids while the DNA fragment did not. Furthermore, one of the three ionic liquids denatured the oligonucleotide duplex. However, both figures 2 and 9 and show little variation of RMSD values during the 5 ns. If the equilibration step was done with frozen atomic coordinates, one would expect to see the original structure at time t=0. If both illustrations show the last 5 ns of each MD simulation, the illustrations should be corrected to accurately reflect the actual time, although the entire MD run should be presented and not only the last 5 ns.

We have chosen to report equal simulation time in Figures 2 and 9. The RMSD portions we show are representative of the equilibrated systems. The RMSD initial growth is very fast and at "point zero" the initial displacement would appear essentially as a vertical line. We have added explaining text in the comments of figure 2 and 9. This is a consequence of the frozen structure initial equilibration as the reviewer has pointed out (not that we had other choices, anyway). In other words, the very first few ps are affected by an artificial strain that dissipates early on. 

  1. Figure 9 caption includes "Figure 1", probably from an earlier version.

We have corrected the error. 

  1. Out of 8 MD simulations, some were 15 ns runs while others were shorter, however it is not clear how long was each distinct simulation. This information is especially important and it should be included in the Methods section.

We have added a section S5 in the SI with the details of the simulations. We have also added a sentence in the methods section to justify our choices. 

  1. On one hand, the initial structure of the oligopeptide is based on NMR, which could account for the lesser variability during the MD simulation. On the other hand, the initial structure of the DNA fragment is based on X-ray and I think it would be worthwhile to compare the NMR and X-ray pdb structures of similar DNA fragments reported in the database. What is the RMSD between these structures?

We are quite sure that our results (especially the denaturation of the oligonucleotide) couldn't be anything different even if we started from an NMR structure in solution. Indeed, in the water "control simulation"  we have only seen minor adjustments with respect to the reference (X-ray) structure. We did a quick search, but we were unable to find NMR structures for short DNA duplexes that could be used in this context. 

  1. One final suggestion: In the conclusions section, the last sentence should be reworded. I understand its meaning, but a minor change could elevate the significance of this work.

We have rephrased it.